# Brain Perfusion Alterations Induced by Standalone and Combined Non-Invasive Brain Stimulation over the Dorsolateral Prefrontal Cortex

**DOI:** 10.3390/biomedicines10102410

**Published:** 2022-09-27

**Authors:** Lais Boralli Razza, Pedro Henrique Rodrigues da Silva, Geraldo F. Busatto, Fábio Luis de Souza Duran, Juliana Pereira, Stefanie De Smet, Izio Klein, Tamires A. Zanão, Matthias S. Luethi, Chris Baeken, Marie-Anne Vanderhasselt, Carlos Alberto Buchpiguel, André Russowsky Brunoni

**Affiliations:** 1Serviço Interdisciplinar de Neuromodulação, Laboratório de Neurociências (LIM-27), Departamento Instituto de Psiquiatria, Hospital das Clínicas, Faculdade de Medicina, Universidade de São Paulo, São Paulo 05403-010, Brazil; 2Department of Head and Skin, Psychiatry and Medical Psychology, Ghent University Hospital, Ghent University, 9000 Ghent, Belgium; 3Ghent Experimental Psychiatry (GHEP) Lab, Ghent University, 9000 Ghent, Belgium; 4Laboratório de Neuroimagem em Psiquiatria (LIM-21), Instituto de Psiquiatria, Faculdade de Medicina da Universidade de São Paulo, R. Dr. Ovidio Pires de Campos 785, São Paulo 05403-000, Brazil; 5Department of Psychiatry (UZBrussel), Free University Brussels, 1090 Brussels, Belgium; 6Department of Electrical Engineering, Eindhoven University of Technology, 5600 MB Eindhoven, The Netherlands; 7Department of Experimental Clinical and Health Psychology, Ghent University, 9000 Ghent, Belgium; 8Divisão de Medicina Nuclear (LIM-43), Instituto de Radiologia, Hospital das Clínicas, Faculdade de Medicina, Universidade de São Paulo, São Paulo 01246-904, Brazil; 9Departamento de Clínica Médica, Faculdade de Medicina da Universidade de São Paulo & Hospital Universitário, Universidade de São Paulo, Av. Prof Lineu Prestes 2565, São Paulo 05508-000, Brazil; 10Hospital Universitário, Universidade de São Paulo, São Paulo 01246-904, Brazil

**Keywords:** non-invasive brain stimulation (NIBS), neuroimaging, Single-Photon Emission Computed Tomography (SPECT), combined interventions, synergistic effect, dorsolateral prefrontal cortex (DLPFC)

## Abstract

Non-invasive brain stimulation (NIBS) interventions are promising for the treatment of psychiatric disorders. Notwithstanding, the NIBS mechanisms of action over the dorsolateral prefrontal cortex (DLPFC), a hub that modulates affective and cognitive processes, have not been completely mapped. We aimed to investigate regional cerebral blood flow (rCBF) changes over the DLPFC and the subgenual anterior cingulate cortex (sgACC) of different NIBS protocols using Single-Photon Emission Computed Tomography (SPECT). A factorial, within-subjects, double-blinded study was performed. Twenty-three healthy subjects randomly underwent four sessions of NIBS applied once a week: transcranial direct current stimulation (tDCS), intermittent theta-burst stimulation (iTBS), combined tDCS + iTBS and placebo. The radiotracer 99m-Technetium-ethylene-cysteine-dimer was injected intravenously during the NIBS session, and SPECT neuroimages were acquired after the session. Results revealed that the combination of tDCS + iTBS increased right sgACC rCBF. Cathodal and anodal tDCS increased and decreased DLPFC rCBF, respectively, while iTBS showed no significant changes compared to the placebo. Our findings suggest that the combined protocol might optimize the activity in the right sgACC and encourage future trials with neuropsychiatric populations. Moreover, mechanistic studies to investigate the effects of tDCS and iTBS over the DLPFC are required.

## 1. Background

Non-invasive brain stimulation (NIBS) is an increasingly used tool for investigative and therapeutic purposes. They are mainly represented by repetitive transcranial magnetic (rTMS) and transcranial direct current electric stimulation (tDCS) [1]. The former uses electromagnetic pulses to induce neuronal depolarization in the regions beneath the stimulating coil [2]. Recently, theta-burst stimulation (TBS), a new method of rTMS, was found to induce faster and stronger changes in cortical excitability [3]. In turn, tDCS produces weak, direct electric currents via electrodes placed over the scalp. The current does not trigger action potentials but can shift synaptic plasticity towards facilitation or inhibition [4].

Due to its neuromodulatory properties, NIBS has been used over the dorsolateral prefrontal cortex (DLPFC), a hub of the central executive network involved in several cognitive and affective processes. Additionally, the DLPFC is functionally interconnected to the subgenual portion of the anterior cingulate cortex (sgACC) [5], and its modulation might affect cognitive performance in healthy people [6], patients with dementia [7] and mood disorders. For instance, the hypoactivity of the DLPFC and the anticonnectivity between the DLPFC and the sgACC have been associated with major depressive disorder [8], and both TBS and tDCS over the left DLPFC seem to be effective interventions for this condition [9,10].

However, despite promising results, two main limitations are still present. First, aggregate meta-analyses show mixed clinical effects for both rTMS and tDCS as monotherapy over the DLPFC, with a modest overall effect size when compared to the placebo. Second, the mechanisms of action of NIBS over the DLPFC have not been completely mapped. In fact, most of the mechanistic evidence derives from studies addressing motor cortex excitability, a brain region that is relatively easier to probe. In this context, investigative methods such as brain imaging and electroencephalography have been used to investigate how NIBS changes DLPFC activity [11]. Notwithstanding, combining NIBS with electroencephalography (EEG) [12] and magnetic resonance imaging (MRI) [13] presents technical challenges that have not been completely solved yet. Another approach is using molecular imaging methods to probe real-time, NIBS-induced changes in neuronal connectivity. These methods measure the concentration of short-lived, radioactive tracers in the investigated tissue. Despite their versatile abilities to map several aspects of brain activity, such as regional cerebral blood flow (rCBF), molecular imaging methods are not as often used to investigate NIBS as other approaches [14].

Here, we aimed to explore NIBS’s effects over the DLPFC and connected regions such as the sgACC using Single-Photon Emission Computed Tomography (SPECT), employing the radiotracer 99m-Technetium-ethylene-cysteine-dimer (99m-TCD). The 99m-TCD SPECT was chosen because it is one of the most common molecular imaging methods used in neuromodulation studies [14]. Using a factorial design, we investigated whether anodal tDCS, iTBS and both interventions combined would change rCBF in the targeted DLPFC and sgACC bilaterally, in comparison to the placebo. Anodal tDCS and iTBS were applied over the neuronavigated left DLPFC. Additionally, rCBF changes in both hemispheres were assessed as bifrontal tDCS was applied. Our primary hypothesis was that, compared to the placebo, the interaction between tDCS and iTBS would increase rCBF in the targeted DLPFC and in the sgACC. Our secondary hypotheses were that tDCS and iTBS alone would also increase rCBF in the targeted DLPFC in comparison to the placebo, but to a lesser extent than the combined protocol. Moreover, we hypothesized that active tDCS protocols would change rCBF in the subregions of the DLPFC in comparison to the other protocols. Our hypotheses were established based on the previous literature, which shows that preconditioning the target area with tDCS may alter the effects of subsequent rTMS protocols in a manner that generates more robust outcomes when compared to the protocols’ use as monotherapy [15], and studies showing that tDCS and iTBS can change brain activity compared to placebo treatments [16,17].

This study can increase knowledge regarding the mechanisms of the interventions of monotherapy and their combination over the DLPFC and open new horizons for treating psychiatric disorders. 

## 2. Materials and Methods

### 2.1. Design

The study was conducted at the Institute of Psychiatry, University of São Paulo, Brazil, with active recruitment taking place from March 2019 to January 2021 (recruitment was stopped between March and October 2020 due to the COVID-19 pandemic). Participants were asked to give written informed consent before the first session. This study was conducted in accordance with the Declaration of Helsinki and was approved by the Ethics Committee at the São Paulo University Hospital das Clínicas (HCFMUSP) and registered on the Plataforma Brasil database (CAAE: 89310918.8.0000.0068).

We used a factorial (2 × 2 design), within-subjects, double-blinded design in which participants were assigned to receive, in a randomized order: (a) sham tDCS plus sham iTBS (placebo), (b) active tDCS plus active iTBS (double active), (c) sham tDCS plus active iTBS (iTBS-only) and (d) active tDCS plus sham iTBS (tDCS-only). The sessions were applied once a week for four consecutive weeks [18]. For more details, the protocol of this study has been published previously [18]. 

Before each session, baseline measurements were acquired. NIBS sessions consisted of active/sham tDCS for 20 min. During the last 9 min of the tDCS session, active/sham TBS was applied concomitantly (Figure 1). The 99mTC-ECD radiotracer was injected intravenously after iTBS onset. The same procedures were conducted for all sessions. 

### 2.2. Participants

Right-handed, 18–45-year-old subjects, with no prior or present psychiatric or neurological disorders and/or clinical diseases, were included. Participants were recruited through media advertisements and flyers distributed across the campus of the University of São Paulo.

Volunteers were screened using a brief online questionnaire. Those who met inclusion criteria underwent an on-site interview with a trained psychologist to check previous or current psychiatric diagnoses based on the DSM-5 [19], the Hamilton Depression Rating Scale (HDRS) [20], the Beck Depression Inventory (BDI) [21] and the Positive and Negative Affect Schedule (PANAS) scales [22]. Exclusion criteria were specific contraindications for NIBS interventions and MRI (e.g., metal implants), smoking (above 10 cigarettes per day), abuse/dependence on other drugs, pregnancy, and use of psychoactive drugs (including antidepressant drugs and benzodiazepines). 

### 2.3. Procedures

Participants visited the hospital on five occasions. During the first visit, an anatomical 3-Tesla MRI acquisition of the brain was performed (General Electric PET/MRI equipment), followed by an MRI-guided neuronavigation (Brainsight, Rogue Resolutions, Inc., Montreal, QC to target the left and right DLPFCs (MNI152 stereotaxic coordinates: −38, +44, +26 and +38, +44, +26, respectively) [8]. Volumetric images were based on a T1-weighted sequence using a 3D fast-field echo pulse sequence with the following parameters: field of view (FOV) of 25.6, repetition time (TR) of 7.7 ms, echo time (TE) of 3.1 ms and 202 slices. The T1-weighted acquisitions were used for the neuronavigation and for partial volume correction of the SPECT images. 

Volunteers were asked to sleep sufficiently, refrain from intense physical activity, not drink any alcoholic and caffeinated beverages, and not take nicotine 24 h prior to the session and to fast for at least 6 h prior to SPECT. 

Injections of the radiotracer 99mTc-ECD using 20 mCi (555 MBq) were administered during the NIBS sessions by a trained nurse, just after iTBS onset as the radiotracer can take up to 30 min to be absorbed by the body. Participants remained seated in a reclining chair until the end of the NIBS session. Up to 35 min after the radiotracer administration, brain imaging was acquired with SPECT equipment (GE—Milwaukee, model 630, Wisconsin, USA) consisting of a dual-head gamma camera with a dedicated collimator for brain studies (Fan Beam; E.CAM; Siemens, Hoffman Estate, IL, USA). SPECT images were acquired every 20 s, at a 3° angle, in a circular orbit (two turns of 360°), using a 128 × 128 matrix. The equipment was calibrated for a photopeak of 140 keV with a symmetric 10% window. The processing was performed using an interactive reconstruction method (OSEM) with a Butterworth filter (cutoff frequency of 0.57 and serial number of 10). All SPECT exams were performed between 10:00 and 13:00 and lasted 45 min.

### 2.4. Interventions

Based on previous large trials with depressive patients [10,23] and cognitive studies [24], tDCS electrodes were positioned over the left (anode) and right (cathode) DLPFC located via neuronavigation. The electrodes were placed in contact with the participants’ skin. tDCS was applied with a current of 2 mA through 25 cm^2^ saline-soaked sponges for 20 min (Neuroconn DC-Stimulator MR device—Neuroconn GmBH, Ilmenau, Germany). The same montage was used for sham tDCS, but the current only had a brief active period of 30 s ramp up and ramp down at the beginning and at the end of the session. Devices were programmed to deliver active or sham stimulation according to an imputed randomized code, allowing for the blinding of both personnel and participants. 

For the iTBS protocol, the TMS coil was positioned over the anode in a 45-degree angle relative to the midline. The stimulation parameters consisted of 54 cycles of 10 triplet bursts with a train duration of 2 s and an interval of 8 s between trains (1620 pulses) at 110% of the resting motor threshold and lasted 8 min and 40 s. The resting motor threshold was established as the minimum TMS intensity necessary to visually yield a motor-evoked potential in the right abductor pollicis brevis muscle in 5 out of 10 successive attempts during relaxation [25]. Both active and sham iTBS protocols were applied using a B65 Active/Placebo Magventure coil that had two identical sides for delivering active or sham stimulation. The coil side was chosen based on randomized codes. This iTBS protocol was chosen because it was proven to be effective in a large depression trial [9].

### 2.5. Images Pre-Processing and Processing

First, the SPECT and the T1-weighted MRI files were converted from DICOM to ANALYSE using the MRICROn software [26]. SPECT neuroimages were co-registered with the T1-MRI of each subject via PMOD™ software version 3.4. A voxel-based partial volume effect correction via the Meltzer method [27] was performed in all SPECT neuroimages to avoid confounding factors in the radiotracer uptake. All images were spatially normalized to the MNI space template using the SPM12 software (Wellcome Trust Centre of Neuroimaging, London, United Kingdom; DARTEL protocol). All SPECT neuroimages were smoothed with an 8 mm Gaussian filter (FWHM). Differences in global cerebral blood flow among scans were accounted for with rCBF normalization, which was achieved proportionally to the global brain activity using proportional scaling. Overall grand mean scaling at a default value of 50 mL/(min/00 mL) was applied to extract raw cerebral blood flow values from each region of interest, which is described in the following section.

### 2.6. Regions of Interest

Our regions of interest were the DLPFC and the sgACC in both hemispheres. Based on our previous studies [28,29], we used the Sallet et al. atlas [30] to parse the dorsal frontal cortex into ten subregions (clusters), which are identified by their corresponding Brodmann areas (BAs). In our investigation, we analyzed seven DLPFC subregions according to the Sallet et al. atlas: cluster 3 (corresponded to BA 9), 4 (BA 10), 5 and 6 (BA 9/46D and BA 9/46V), 6, 7 (BA 46), 8 (BA 8A) and 10 (BA 8B) (Appendix A). The Sallet et al. atlas also incorporates motor and premotor areas, but they were not included in our analysis as they were not part of our hypotheses. For our primary outcome, we used clusters 5 and 6 together (region on interest (ROI) 9/46D, 9/46V) for both hemispheres, which corresponded to the neuronavigated targets used in the current study (MIN coordinates: −38, 44, 26 and 38, 44, 26, respectively) (Appendix A). For the sgACC, we used an 8 mm sphere centered on the MNI coordinates 6, 16 and −10, as used in Fox et al., 2012 [8].

### 2.7. Statistical Analysis 

The statistical analysis was performed using R software, version 4.1.2 [8,31]. The package lme4 was used to run linear mixed models (LMM) analyses to investigate the association between rCBF and the stimulation group. In the LMM models, we fitted the rCBF of each region of interest as a dependent variable, while *tDCS* (i.e., tDCS yes/tDCS no), *iTBS* (i.e., iTBS yes/iTBS no) and their interaction (*iTBS*tDCS*) were added as fixed effects. The variable ‘subject’ was included as a random effect. The primary investigated regions were the bilateral cluster BA9/46D plus BA9/46V and the bilateral sgACC, which were evaluated in four separate LMMs. For the secondary outcome, we investigated the subregions of the DLPFC. This approach resulted in 7 additional models that were carried out for both hemispheres. For this analysis, the correction for multiple comparisons was carried out using the Bonferroni method, and only the corrected *p*-values are reported here. All models were controlled for effects of age and gender. All results were considered significant at a *p*-value ≤ 0.05. 

Finally, the software CONN [32] was used to create 3D representations of the main regions of interest of our study (bilateral BA9/46V, 9/46D and sgACC). Additional 3D representations were also created for regions of interest that showed significance for the secondary outcome. 

## 3. Results 

Out of the 211 subjects interested in participating, 56 underwent screening with a trained psychologist and 25 were included in our study. However, two participants were excluded after MRI acquisition due to the presence of neurological abnormalities in the exams. Therefore, a total of 23 subjects participated in the current study, and 92 brain imaging assessments were performed (Appendix A). Overall, participants had a mean age of 28.7 (standard deviation (SD) = 7) years and a mean number of years in education of 17.1 (SD = 3.1), and 66.6% were women. 

### 3.1. Primary Outcome

For the primary outcome, tDCS and iTBS main effects were not observed in any analysis. Additionally, we found no significant effects of the interaction between iTBS and tDCS on the left and right 9/46D, 9/46V regions (Figure 2A), but a significant interaction in the right sgACC was observed. Results from the left sgACC show no significance (Table 1; Figure 2B).

No demographic characteristics and baseline measures were found to significantly influence the results (Appendix A). 

### 3.2. Secondary Outcomes for DLPFC Subregions 

For the left DLPFC subregions, tDCS was associated with a significant decrease in cerebral blood flow in the BA8a region (b = −28.73, 95% confidence interval [CI] = −46.26; −11.2, *p* = 0.01). Regarding the right DLPFC, results show that tDCS increased cerebral blood flow in the BA10 region (b = 17.7, 95% CI = 5.2; 30.3, *p* = 0.04) (Figure 3). 

No significant effects were found for the iTBS and the interaction of iTBS and tDCS. 

## 4. Discussion 

We investigated the standalone and combined NIBS protocols on the rCBF of the DLPFC and sgACC in twenty-three healthy participants. To our best knowledge, this is the first study evaluating the interaction effects of two NIBS interventions over the DLPFC using neuroimaging data. Our primary outcome demonstrated increased rCBF over the right sgACC during the interaction of tDCS and iTBS, while the secondary outcome showed a decrease in rCBF in the left DLPFC and an increase in the right DLPFC after active tDCS.

Recent studies suggest that the DLPFC and the right sgACC are functionally correlated during depressive episodes, and changes in their connectivity are associated with higher antidepressant efficacy for rTMS treatments [8,33,34]. Neither rTMS nor tDCS can reach the sgACC region directly, but a connectivity-based study by Fox and colleagues [8] showed that NIBS effects on the right sgACC depend on which specific location is targeted within the left DLPFC. Effects can be maximized using the MNI coordinates −38, +44 and +26, which were used in our study [8]. As we have hypothesized, the interaction between tDCS and iTBS significantly increased cerebral perfusion on the right sgACC in comparison to the placebo.NIBS methods are considered state-dependent interventions, i.e., the state of the target region can influence the overall effects of the intervention [35]. Therefore, in our study, preconditioning the DLPFC with tDCS may have controlled for the underlying ongoing neuronal activity, thereby enhancing the effectiveness of the rTMS stimulation and ultimately allowing their combined effects to reach the sgACC [35]. Moreover, previous studies probing the motor cortex also demonstrated that tDCS priming can increase rTMS effects driving mainly by metaplasticity phenomenon [15,36]. We believe our findings are important for two main reasons. First, they provide further evidence for the strong connection of the DLPFC site identified by Fox and colleagues [8] with the sgACC and extend their connectivity-based findings with measures of rCBF in the sgACC. Second, we possibly identified a potential surrogate marker for the improvement of the clinical response of rTMS and tDCS. Optimizing the functional activity between the DLPFC and the sgACC can be important for the improvement of both mood and cognitive disorders. For instance, the response rates for both rTMS and tDCS treatments are still around 30 to 40% for depression [37], and the combination of NIBS interventions might provide an option to optimize clinical gains. Moreover, as the sgACC also plays an important role in cognitive impairment disorders [38], future trials are encouraged to investigate the combined protocol as a potential treatment for these neuropsychiatric conditions. 

Moreover, we found that tDCS decreased and increased rCBF over the left and right DLPFC, respectively. As we positioned the anode over the left and the cathode over right hemispheres, our results challenge initial findings that indicate anodal stimulation leads to an increase in cortical excitability, while cathodal stimulation leads to a decrease [4]. However, recent studies suggest that cathodal intensities higher than 1 mA are associated with motor cortical facilitation, instead of inhibition [39,40]. A 2 mA cathodal stimulation leading to an rCBF increase is in line with this literature. Moreover, spectroscopy studies from Hone-Blanchet and colleagues [41] and Mezger and colleagues [42] showed that 1 mA bilaterally applied over the DLPFC increased the N-Acetylaspartate level over the left PFC, while 2 mA did not show any metabolic changes. Considering that non-linear dose-dependent effects of tDCS using lower and higher intensities were also observed over the left PFC of healthy subjects evaluated by cognitive outcomes [43], we might consider that our findings can be explained not only, but also by the non-linear effects of tDCS using 2 mA. 

Finally, our findings show that iTBS alone did not change cerebral brain perfusion in comparison to the placebo in both DLPFC and sgACC regions. This is interesting because even though iTBS is a newer rTMS protocol, neuroimaging studies show one session of iTBS significantly changed metabolic and activity measures [17]. Moreover, a recent study found that 1200 pulses of iTBS decreased motor cortical excitability, while protocols with a lower (600) or higher (1800) number of pulses did not lead to changes compared to the sham [44]. Accordingly, our non-significant findings could be explained by the fact that we used an intense iTBS protocol of 1620 pulses, which was closer to the 1800 pulses that were found to be ineffective than to the 1200 pulses that were found to be effective.

This study has some limitations that should be underscored. First, the sample size was small and may have been insufficient to clearly delineate differences across interventions. Second, voxelwise analyses were not performed. However, we chose ROI-based analyses because the averaging over voxels might help to limit results based on singular extreme values, which is a typical problem associated with small-sample studies. Third, it is important to acknowledge that SPECT is a neuroimaging technique that presents poorer temporal and spatial resolution in comparison to other techniques, such as Positron Emission Tomography (PET) and functional magnetic resonance imaging (fMRI), which could have been more sensitive to the brain functional changes. However, compared with PET, SPECT uses longer-lived, more easily obtained radioisotopes. Furthermore, SPECT allowed us to inject the radiotracer during TBS and tDCS stimulation and thus to evaluate acute changes in brain perfusion, which would have been very challenging to achieve with fMRI.

## 5. Conclusions

In the present study, we conducted a factorial, within-subject, double-blinded trial, in which participants were randomly assigned to receive four NIBS sessions (tDCS alone, iTBS alone, tDCS + iTBS and a placebo), applied once a week. The results show that combining tDCS and iTBS over the neuronavigated DLPFC of healthy participants was associated with increased activity in the right sgACC, but not in the targeted DLPFC region. However, tDCS alone led to an increase in rCBF under the cathodal DLPFC simulation site and decrease under the anodal DLPFC simulation. iTBS did not show rCBF changes in comparison to the placebo. Our findings suggest that the combination of tDCS and iTBS over the neuronavigated left DLPFC can optimize the activity in the right sgACC, while tDCS using 2 mA might lead to non-linear effects in the DLPFC. These findings encourage future trials evaluating the combination of tDCS and iTBS with neuropsychiatric populations and mechanistic studies to evaluate the effects of 2 mA tDCS and 1600 pulses of iTBS over the DLPFC, as was used in this study.

## Figures and Tables

**Figure 1 biomedicines-10-02410-f001:**
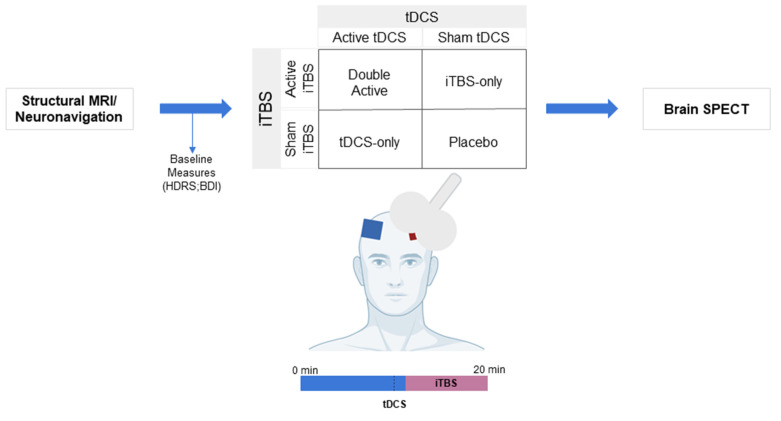
Study design. Structural MRI and the neuronavigation were performed in the first session. Afterwards, participants underwent four different NIBS sessions on different days. TDCS and iTBS (active or sham) were applied simultaneously in all sessions. Abbreviations: BDI: Beck Depression Inventory; HDRS: Hamilton Depression Rating Scale; iTBS: Intermittent theta-burst stimulation; min: minutes; MRI: magnetic resonance imaging; SPECT: Single-Photon Emission Computed Tomography; tDCS: transcranial direct current stimulation.

**Figure 2 biomedicines-10-02410-f002:**
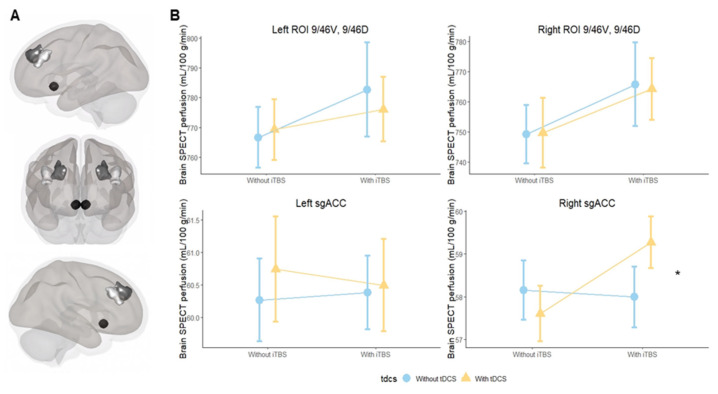
(**A**). Main regions of interest; (**B**). Cerebral blood flow changes in NIBS protocols over the targeted DLPFC and the sgACC. Note: Region BA 9/46D is represented by dark gray, region BA 9/46V is represented by light gray, and the sgACC is represented by black. Abbreviations: iTBS: intermittent theta-burst stimulation; min: minutes; mL: milliliter; g: gram; SPECT: Single-Photon Emission Computed Tomography; sgACC: subgenual anterior cingulate cortex; tDCS: transcranial direct current stimulation; *: significant result.

**Figure 3 biomedicines-10-02410-f003:**
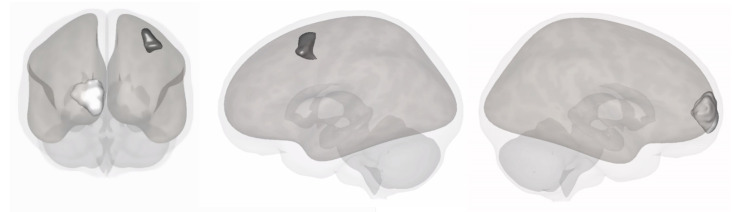
DLPFC regions that presented a significant regional cerebral blood flow change after tDCS.

**Table 1 biomedicines-10-02410-t001:** Main results. Abbreviations: CI: confidence interval; iTBS: intermittent theta-burst stimulation; sgACC: subgenual anterior cingulate cortex; tDCS: transcranial direct current stimulation. Note: all active interventions were compared to sham.

	Left Hemisphere				Right Hemisphere			
	Beta	95% CI		*p*-Value	Beta	95% CI		*p*-Value
**iTBS vs. tDCS**								
9/46D, 9/46V	−9.17	−40.04	21.7	0.55	−2	−33.35	29.3	0.9
sgACC	−0.4	−2.4	1.7	0.7	**19**	**1.69**	**36.1**	**0.03**
**tDCS**								
9/46D, 9/46V	2.58	−19.25	24.4	0.8	0.5	−21.7	22.7	1
sgACC	0.47	−1	1.9	0.52	−7.4	−19.6	4.8	0.2
**iTBS**								
9/46D, 9/46V	16.01	−5.82	37.4	0.15	16.6	−5.6	38.7	0.14
sgACC	0.11	−1.35	1.5	0.9	−7.3	−19.5	4.9	0.2

## Data Availability

Not applicable.

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
