# Peer review of "Brain Perfusion Alterations Induced by Standalone and Combined Non-Invasive Brain Stimulation over the Dorsolateral Prefrontal Cortex"

_biomedicines, 2022, doi:10.3390/biomedicines10102410_

Round 1
Reviewer 1 Report
The authors deal with a relevant and timely, i.e., the brain perfusion alterations induced by standalone and combined non-invasive brain stimulation (NIBS) over the dorsolateral prefrontal cortex (DLPFC), which was approached through a relatively novel approach. Namely, they aimed at investigating the regional cerebral blood flow (rCBF) changes over the DLPFC and the interconnected subgenual anterior cingulate cortex (sgACC) of different NIBS protocols using Single Photon Emission Computed Tomography. Briefly, 23 healthy subjects randomly underwent 4 sessions of navigated NIBS once a week: transcranial direct current stimulation (tDCS), intermittent theta-burst stimulation (iTBS), combined tDCS+iTBS, and placebo. Results revealed that none of the active interventions changed rCBF on the targeted DLPFC, but the combination of tDCS+iTBS increased rCBF on the right sgACC. Also, tDCS increased and decreased rCBF on DLPFC subregions under the cathode and anode, respectively, while iTBS did not show significant changes.
Overall, the study is nicely conceived and designed; the results seem to be consistent and are adequately presented. Few comments to the authors, requiring some revision.
Abstract: the results obtained are not discussed; please shortly mention them, also highlighting their translational value and potential clinical implications in both mood and cognitive disorders. In addition, please specify the double-blind design of the study also in this section.
Introduction: among the increasing applications of NIBS, those that seem to be relevant to this article are related to their use in cognition, both in normal conditions (i.e., “self-neuroenhancement”) and in patients with a variety of cognitive disorders (for recent papers, please see PMID: 31942774 and PMID: 34482205, respectively). Additionally, at the end of this section, please clearly state the experimental design of the study: what did the authors expect? How? Why?
Methods: given the effect of both nicotine and caffeine on brain excitability and rCBF, also light smoking (<10 cigarettes per day) and coffee intake (at least prior to the examinations) should be included among the exclusion criteria. More importantly, please provide the rationale and supporting reference(s) underlying the NIBS protocol adopted, and in particular the choice of stimulate for 4 sessions once a week (that, globally, seems to be a rather weak stimulation protocol). Finally, please provide more technical and procedural details on both NINS techniques adopted (tDCS and TBS), including the current international guidelines followed for their use (e.g., PMID: 27866120 and PMID: 25797650, respectively).
Results: current figures do not seem to be at high resolution; if possible, please provide clearer images.
Discussion: as mentioned above, the results obtained are not deeply discussed; please further elaborate on them, also highlighting their translational value and potential clinical implications in both mood and cognitive disorders. Also, among the potential mechanisms underlying the effect of NIBS techniques, please mention the induction and modulation of metaplasticity and related phenomena (e.g., PMID: 34276553).
References: please check the citation list for completeness and accuracy (e.g., n. 2, 14, and 23).
General: although the language is overall acceptable, an editing by a native-English speaker will be useful.
Reviewer 2 Report
This study aimed to investigate NIBS effects over the DLPFC and interconnected re-gions such as the sgACC using Single Photon Emission Computed Tomography (SPECT), employing the radiotracer 99m-Technetium-ethylene-cysteine-dimer (99m-TCD). The 99m-TCD SPECT was chosen because it is one of the most common molecular imaging methods used in neuromodulation studies to date. Therefore, the authors investigated, using a factorial design, whether anodal tDCS, iTBS and both interventions combined, would change rCBF in the targeted DLPFC and sgACC bilaterally, compared to placebo. Anodal tDCS and iTBS were applied over the neuronavigated left DLPFC. Notwithstanding, rCBF changes in both hemispheres were assessed, as bifrontal tDCS was applied. This study can increase knowledge regarding the mechanisms of the interventions of monotherapy and their combination over the DLPFC and open new horizons for treating psychiatric disorders.
The manuscript is well structured and the study is of great interest to the scientific and current community. I have just a few suggestions for the authors.
In the caption of figure 1, before specifying the acronyms used, it would be appropriate to insert a title that describes the figure, as rightly done in figure 2.
Table 1 is difficult to understand for readers. If possible it should be reorganized to facilitate understanding
In the discussion paragraph I believe there was an editing error regarding the insertion of references in the test. For example the authors have written some references like this: [6,25], [26]; [30] [31]. These details should be corrected by following the guidelines for references in the guidelines for authors.
If possible, the paragraph on conclusions should be expanded a little.
Author Response
请参阅附件。
